# Metal and Phosphate Ions Show Remarkable Influence on the Biomass Production and Lipid Accumulation in Oleaginous *Mucor circinelloides*

**DOI:** 10.3390/jof6040260

**Published:** 2020-10-30

**Authors:** Simona Dzurendova, Boris Zimmermann, Valeria Tafintseva, Achim Kohler, Svein Jarle Horn, Volha Shapaval

**Affiliations:** 1Faculty of Science and Technology, Norwegian University of Life Sciences, Drøbakveien 31, 1430 As, Norway; boris.zimmermann@nmbu.no (B.Z.); valeria.tafintseva@nmbu.no (V.T.); achim.kohler@nmbu.no (A.K.); volha.shapaval@nmbu.no (V.S.); 2Faculty of Chemistry, Biotechnology and Food Science, Norwegian University of Life Sciences, Christian Magnus Falsens vei 1, 1433 As, Norway; svein.horn@nmbu.no

**Keywords:** *Mucor circinelloides*, high-throughput screening, metal ions, phosphorus, lipids, biofuel, FTIR spectroscopy, bioremediation, co-production

## Abstract

The biomass of *Mucor circinelloides*, a dimorphic oleaginous filamentous fungus, has a significant nutritional value and can be used for single cell oil production. Metal ions are micronutrients supporting fungal growth and metabolic activity of cellular processes. We investigated the effect of 140 different substrates, with varying amounts of metal and phosphate ions concentration, on the growth, cell chemistry, lipid accumulation, and lipid profile of *M. circinelloides*. A high-throughput set-up consisting of a Duetz microcultivation system coupled to Fourier transform infrared spectroscopy was utilized. Lipids were extracted by a modified Lewis method and analyzed using gas chromatography. It was observed that Mg and Zn ions were essential for the growth and metabolic activity of *M. circinelloides.* An increase in Fe ion concentration inhibited fungal growth, while higher concentrations of Cu, Co, and Zn ions enhanced the growth and lipid accumulation. Lack of Ca and Cu ions, as well as higher amounts of Zn and Mn ions, enhanced lipid accumulation in *M. circinelloides.* Generally, the fatty acid profile of *M. circinelloides* lipids was quite consistent, irrespective of media composition. Increasing the amount of Ca ions enhanced polyphosphates accumulation, while lack of it showed fall in polyphosphate.

## 1. Introduction

*Mucor circinelloides* is a dimorphic oleaginous filamentous fungus with a fully sequenced genome [1]. It has a versatile metabolism, allowing utilization of a variety of feedstocks, making this fungus widely applicable in a range of biotechnological processes [2]. *M. circinelloides* is well known as a robust cell factory, where extracellular products include enzymes (cellulases, lipases, proteases, phytases, and amylases) [2,3] and ethanol [4]. Further, *M. circinelloides* can synthesize and accumulate a number of valuable intracellular components, such as lipids, polyphosphates, carotenoids, and chitin/chitosan [5,6,7,8,9]. The biomass of *M. circinelloides* has a significant nutritional value and can be used as a feed ingredient [10]. Chitosan exhibits great chelating properties, mainly due to the low level of acetylation and the abundance of hydroxyl groups [9,11,12]. Due to the presence of chitin and chitosan in the cell wall of *M. circinelloides*, the biomass of this fungus can be used as bioabsorbent for heavy metals and applied as a bioremediation agent, for example in the wastewater treatment [13].

*M. circinelloides* has been extensively studied for the production of lipids for different applications [14,15,16]. The lipids are mainly in the form of triacylglycerides (TAGs) and contain palmitic, stearic, oleic, linoleic acid, and y-linoleic acids that make it particularly suitable for biodiesel production [17]. Therefore, the biomass of *M. circinelloides* could be considered as an important alternative feedstock for the biodiesel industry [18]. However, the cost of the *M. circinelloides* biomass production for biodiesel as a sole product is still too high compared to competitive bioprocesses. Thus, there is a need to further optimize lipid and biomass yield for *M. circinelloides*, and develop a coproduction concept, where other valuable components could be produced along with lipids in a single fermentation process [19,20].

Lipid accumulation, as well as biomass formation, can be affected by many different cultivation parameters, such as the nutrient composition of the growth medium, temperature, pH, aeration, or parameter shift during the fermentation (temperature/pH) [2,21,22]. Optimization of the nutrient composition of the growth medium is one of the most important aspects in improving fungal lipid production. In order to increase lipid and biomass yields, it is crucial to understand the role and effect of all media components. Many studies have assessed the effect of different carbon (C) and nitrogen (N) sources on the fungal lipid production in *Mucoromycota* fungi [23,24,25,26,27,28,29], where nitrogen limitation in carbon-rich media is the most frequently used strategy for inducing lipid accumulation and achieving high lipid yields. Macro- and micronutrients, such as phosphorus (P), potassium (K), sulfur (S), calcium (Ca), sodium (Na), iron (Fe), and magnesium (Mg), have previously been reported as essential for optimal fungal growth and metabolic activity [30].

Metal ions play an important role in fungal metabolism as they provide necessary redox and catalytic activities for the cellular processes [31]. The role of metal ions in yeast metabolism has been widely studied [32,33,34,35]. Bivalent metal ions are often reported as cofactors for different enzymes [36,37]. Metal ions are usually examined in the context of bioremediation capabilities of *M. cicrcinelloides* [9,38], while the role of metal ions in lipid accumulation of *Mucoromycota* fungi have been examined only to a limited extent. Different metal ions have shown strain-specific influence on lipid accumulation in *Mucoromycota* fungi, where either fatty acid profiles or total lipid content is affected. For example, manganese (Mn) has shown positive effects on the lipogenesis in *Mucor plumbeus* and *Mortierella* sp. [39,40]. Iron had an inhibiting effect on the arachidonic acid production in *Mortierella* sp. [39], while together with magnesium and zinc (Zn), it was enhancing lipid accumulation in *Cunninghamella* sp. Furthermore, zinc increased the gamma-linoleic acid production in *Cunninghamella* sp. [41], while iron, zinc, and copper (Cu) were reported as enhancers of arachidonic acid production in *Mortierella alpina* [42]. To the authors’ knowledge, the effect of calcium and cobalt (Co) on the lipid accumulation in the oleaginous *Mucoromycota* fungi has not been investigated before. Moreover, there have been no studies reporting the effect of metal ions on the lipogenesis in *M. circinelloides*.

In our previous studies, we have assessed the chemical composition of nine different oleaginous filamentous fungi (including *M. circinelloides*) and revealed nutrient-induced coproduction of lipids, chitin/chitosan, and polyphosphates [43,44]. We reported that *M. circinelloides* has an ability to coproduce lipids, polyphosphate, and chitin/chitosan at different phosphorus concentrations and showed a versatile metabolism with a high adaptability level to different stress conditions. Thus, this fungus can be utilized in phosphorus recovery processes, while the co-production concept greatly contributes to the economic feasibility of such processes.

The aim of this study was to assess the effect of 140 different substrates, with varying amounts of metal and phosphate ions, on the growth, cell chemistry, and lipid production in *M. circinelloides.* Different concentrations of phosphorus source were used in order to study the effect of metal ions on the co-production of lipids, polyphosphates and cell wall polysaccharides, such as chitin/chitosan, triggered by phosphorus availability. Analogous to our previous studies, the study was performed in a high-throughput set-up using a Duetz microtiter plate system (Duetz-MTPs) combined with Fourier transform infrared (FTIR) spectroscopy [43,44,45,46]. FTIR spectroscopy was applied to obtain a biochemical fingerprint of the microbial cells [43,47,48,49,50,51,52], while gas chromatography was used to analyze the lipid yield and fatty acid profiles of the extracted lipids.

## 2. Materials and Methods

### 2.1. Growth Media and Cultivation Conditions

Fungal strain *M. circinelloides* VI04473, provided by the Veterinary Institute, Oslo, Norway was selected based on the previous study of 100 oleaginous fungal strains, as it showed the highest lipid and biomass yield of all tested *Mucor* strains [50]. Moreover, this strain has also shown coproduction potential for lipids, chitin/chitosan, and polyphosphates [43]. *M. circinelloides* was cultivated on malt extract agar (MEA) for 7 days at 25 °C in order to obtain fresh spores for the inoculation into nitrogen-limited broth media with various metal and phosphorus ion concentrations. Spores were collected from agar plates using 10 ml of saline solution and a bacteriological loop. The composition of the reference medium, used and modified in our previous studies [27,43,44,45], was the following: 80 g/L of glucose, 1.5 g/L of (NH_4_)_2_SO_4_, 7 g/L of KH_2_PO_4_, 2 g/L of Na_2_HPO_4_, 1.5 g/L of MgSO4·7H2O, 0.1 g/L of CaCl2·2H2O, 0.008 g/L of FeCl3·6H2O, 0.001 g/L of ZnSO4·7H2O, 0.0001 g/L of CoSO4·7H2O, 0.0001 g/L of CuSO4·5H2O, 0.0001 g/L of MnSO4·5H2O, where the listed concentrations of the metal ions Ca, Cu, Co, Fe, Mg, Mn, and Zn were assigned as reference concentration and marked as “R” (Table 1). The higher—1000; 100; 10 × R and lower—0.1; 0.01 and 0 × R concentrations of the metal ions were assessed in the study as described in Table 1. The reference medium was modified by using four relative levels of metal and phosphate ions (Table 1). KH_2_PO_4_ and Na_2_HPO_4_ were used as phosphates substrate, and their total concentration is hereafter referred as “phosphates concentration” (Pi). Phosphate concentrations KH_2_PO_4_ 7 g/L, Na_2_HPO_4_ 2 g/L have been assigned as Pi1. In addition to Pi1 concentration, the higher—4 and 2 × Pi1 and lower—0.5 and 0.25 × Pi1 concentrations of phosphates were assessed in the study as described in Table 1. Broth media, with the lower than Pi1 amount of Pi—Pi0.5 and Pi0.25, contained KCl and NaCl in a corresponding concentration in order to have equal amounts of K^+^ and Na^+^ ions as in the Pi1 condition. Different media were prepared by modifying one metal ion concentration at a time for every level of Pi ions. The only exception was for the condition 0Mg10Ca, which was tested in order to examine a possibility of substitution of Mg by Ca. Thus, in total, 140 different media were prepared. The concrete concentrations of all media components can be found in the Appendix A.

Considering the high-throughput set-up of the study and consequently high number of samples, the reproducibility of the *M. circinelloides* growth was controlled by four biological replicates for the reference medium R with Pi1 concentration, two biological replicates for the reference metal ion concentrations, and the following phosphate ion concentrations: Pi4, Pi2, Pi0.5, and Pi0.25, and two biological replicates for the medium with 0Ca under all tested phosphate concentrations. The reproducibility of lipid accumulation was controlled under the reference metal concentrations by four biological replicates for the reference medium R-Pi1, two biological replicates for Pi4, Pi2, Pi0.5, and Pi0.25 media, and two biological replicates for the media with 0Ca-Pi2 and Pi1 levels. The biological variability is represented by error bars in Figure 1 and standard deviation in Table 3.

Cultivation was performed in Duetz-MTPS (Enzyscreen, Heemstede, The Netherlands) [43,45,47,50,53,54], consisting of 24-square well polypropylene deep well microtiter plates (MTPs), low evaporation sandwich covers with a clamp system. A total of 7 ml of sterile media broth was transferred into the autoclaved microtiter plates and each well was inoculated with 50 µL of the spore suspension. MTPs were placed on the shaking platform of the incubator MAXQ 4000 (Thermo Scientific, Oslo, Norway). Cultivations were performed for 7 days at 25 °C and 400 rpm agitation speed (1.9 cm circular orbit).

### 2.2. Lipid Extraction and GC-FID Analysis of Lipid Concentration and Fatty Acid Profile

Direct transesterification was performed according to Lewis et al. [55], with some modifications [44]: 2 mL screw-cap polypropylene (PP) tubes were filled with 30 ± 3 mg of freeze-dried biomass, 250 ± 30 mg of acid-washed glass beads, and 500 μL of methanol. Further, the fungal biomass was disrupted in a tissue homogenizer (Bertin Technologies Percellys Evolution, Montigny-le-Bretonneux, France). The disrupted fungal biomass was transferred into glass reaction tubes by washing the PP tube with 2400 μL of a methanol–chloroform–hydrochloric acid solvent mixture (7.6:1:1 *v/v*). Then, 1 mg of C13:0 TAG internal standard in 100 µL of hexane was added to the glass reaction tubes (100 µL from a 10.2 mg/mL^−1^ glyceryl tritridecanoate (C_42_H_80_O_6_, C13:0 TAG (13:0/13:0/13:0), Sigma-Aldrich, St. Louis, Missouri, USA). Reaction tubes were incubated at 90 °C for 1 h, followed by cooling to room temperature and the addition of 1 mL distilled water. The fatty acid methyl esters (FAMEs) were extracted by the addition of 2 mL of a hexane–chloroform mixture (4:1 *v/v*) and applying 10 s of vortex mixing. The reaction tubes were centrifuged at 3000 g for 5 min at 4 °C and the upper hexane phase was collected in glass tubes. The extraction step was repeated three times for each sample. Subsequently, the solvent was evaporated under nitrogen at 30 °C and FAMEs were dissolved in 1.5 mL of hexane containing 0.01% of butylated hydroxytoluene (BHT, Sigma-Aldrich, St. Louis, Missouri, USA) and a small amount of anhydrous sodium sulfate (to remove traces of water in the sample). Samples were mixed by vortexing and, finally, dissolved FAMEs were transferred to the GC vials.

Fatty acid profile analysis was performed using a gas chromatography system with flame ionization detector (GC-FID) 7820A GC System, Agilent Technologies, controlled by Agilent OpenLAB software (Agilent Technologies, Santa Clara, CA, USA). Agilent J and W GC column 121–2323, DB-23, 20 m length; 0.180 mm diameter; 0.20 µm film was used for the separation of FAMEs. Then, 1 µL of the sample was injected in the 30:1 split mode with the split flow 30 mL/min. The inlet heater temperature was set on 250 °C and helium was used as the carrier gas. The flow of helium through the column was 1 mL/min. The total runtime for one sample was 36 min with the following oven temperature increase: initial temperature 70 °C for 2 min, after 8 min to 150 °C with no hold time, 230 °C in 16 min with 5 min hold time, and 245 °C in 1 min with 4 min hold time. For identification and quantification of fatty acids, the C4–C24 FAME mixture (Supelco, St. Louis, MO, USA) was used as an external standard, in addition to C13:0 TAG internal standard. The weight of individual FAs was calculated based on the peak areas, relative response factors (RRF), and C13 internal standard. The total lipids in the fungal biomass were the sum of FA (the weight of C13 IS was subtracted) divided by the weight of dry biomass.

### 2.3. Fourier Transform Infrared Spectroscopy of Fungal Biomass

Fourier transform infrared (FTIR) spectroscopy analysis of fungal biomass was performed according to Kosa et al. [45], with some modifications [43]. The biomass was separated from the growth media by centrifugation and washed with distilled water. Approximately 5 mg of washed biomass was transferred into a 2 ml polypropylene tube containing 250 ± 30 mg of acid washed glass beads and 0.5 ml of distilled water for further homogenization. The remaining washed biomass was freeze-dried for 24 h for determining biomass yield. The homogenization of fungal biomass was performed by using Percellys Evolution tissue homogenizer (Bertin Technologies, Aix-en-Provence, France) with the following set-up: 5500 rpm, 6 × 20 s cycle. Then, 10 µL of homogenized fungal biomass was pipetted onto an IR transparent 384-well silica microplate. Samples were dried at room temperature for 2 h. In total, 140 biomass samples were analyzed in three technical replicates by FTIR spectroscopy.

FTIR spectra were recorded in a transmission mode using the high throughput screening extension (HTS-XT) unit coupled to the Vertex 70 FTIR spectrometer (both Bruker Optik, Leipzig, Germany). Spectra were recorded as the ratio of the sample spectrum to the spectrum of the empty IR transparent microplate in the region between 4000 cm^−1^ and 500 cm^−1^, with a spectral resolution of 6 cm^−1^, a digital spacing of 1.928 cm^−1^, and an aperture of 5 mm. For each spectrum, 64 scans were averaged. In total, 420 biomass spectra were obtained.

The OPUS software (Bruker Optik GmbH, Leipzig, Germany) was used for data acquisition and instrument control.

### 2.4. Data Analysis 

The following software packages were used for the data analysis: Unscrambler X version 10.5.1 (CAMO Analytics, Oslo, Norway) and Matlab R2019a (The Mathworks Inc., Natick, MA, USA).

#### 2.4.1. Analysis of FTIR Spectral Data

To evaluate the correlation of lipid content results obtained from FTIR and GC data and, further, investigate media associated changes of the lipid content in biomass, FTIR spectra were preprocessed by 2nd derivative using Savitzky–Golay algorithm with 2nd order polynomial and windows size 13, followed by spectral region of interest (SROI) selection 3050–2800 and 1800–1700 cm^−1^ and normalization by extended multiplicative signal correction (EMSC) [56] with linear and quadratic terms. Preprocessed FTIR data were then analyzed by principal component analysis (PCA) and score plots were used to compare the lipid related information in FTIR and GC data.

Different preprocessing was applied in order to evaluate the cell chemistry changes that occurred under different media conditions. To do so, SROI 3300–2800 and 1800–800 cm^−1^ was selected and normalized by EMSC using linear and quadratic terms and up-weighting the region 2800–1800 cm^−1^. The up-weighting of the inactive region 2800–1800 cm^−1^ helped in reducing baselines by EMSC in the SROI. Afterwards, the dataset was split according to the concentrations of inorganic phosphorus (Pi) and separately analyzed by PCA. Correlation loading plots were obtained for Pi1, Pi2, and Pi4 in order to analyze the most pronounced correlation patterns in the data. To obtain such a plot, we used scores of each separate PCA model corresponding to one Pi concentration and projected on them variables of interest such as certain relevant peaks of FTIR data and other reference variables, such as pH, biomass yield, and lipid content from GC data, in addition to experimental design factors. The maxima of the corresponding chemical bonds selected for correlation loading plots based on the FTIR spectra of reference materials were pure sodium polyphosphate, chitin, and glyceryl trioleate (Table 2, Appendix A). These compounds were the main cell components of interest. For plotting the peaks on the correlation loading plots, the preselected peaks of the preprocessed spectra were used (Table 2).

#### 2.4.2. Analysis of GC Data 

The detailed fatty acid profiles from the GC analysis were analyzed and compared to FTIR data. Fatty acid profiles were represented by both single fatty acids and sum of fatty acids—saturated fatty acids (SAT), monounsaturated fatty acids (MUFA), and polyunsaturated fatty acids (PUFA). Each data column was standardized (x/std (x)) and then analyzed by PCA. The scatter plot was used to compare the information related to total lipid content and profile obtained by GC and FTIR.

To find differences in fatty acid composition at different Pi concentrations, after standardization, the GC dataset was split into three datasets: Pi1, Pi2, Pi4, and analyzed separately by PCA. To learn about the correlation patterns in the data, the correlation loading plots for GC data were obtained using scores of GC based PCA models. The fatty acid profile of *M. circinelloides* is dominated by myristic (C14:0), palmitic (C16:0), palmitoleic (C16:1), stearic (C18:0), oleic (C18:1n9), linoleic (C18:2n6), and γ-linolenic (C18:3n6) acid, therefore these were presented in the correlation loading plots.

## 3. Results

### 3.1. Growth of Oleaginous M. Circinelloides in Metal Ion-Regulated Media with Different Pi Levels

In order to assess the variation between bioreplicates, cultivation of *M. circinelloides* in the reference medium R with Pi1 concentration was performed 11 times (part of the data were published previously [43]). Cultivations were performed at different timepoints and by using different MTPs. The results of statistical analysis of 11 bioreplicates show that the mean for the biomass production in the reference medium (R-Pi1) was 10.08 g/L (range: 9.35–11.52 g/L, median: 9.90 g/L), with 0.62 g/L standard deviation, and 0.19 g/L standard error. Therefore, it can be considered that all deviations higher than two standard deviations (13% of the average biomass concentration) are statistically significant and can be assigned as the effect of various metal and phosphates concentrations.

Growth of *M. circinelloides* in different media was strongly affected by the availability of phosphates. Low availability of phosphates (Pi0.25 and Pi0.5) led to a substantial decrease in pH (Appendix A) causing a significantly reduced growth of *M. circinelloides* (Figure 1, Appendix A). Biomass yields in all Pi0.25 and Pi0.5 media were in the range from 0.51 g/L to 2.79 g/L and from 0.73 g/L to 4.70 g/L, respectively (Figure 1). The biomass production under moderate and high levels of phosphates (Pi1–Pi4) was substantially higher and in the range from 7.84 g/L to 12.90 g/L for Pi1, from 7.54 g/L to 12.47 g/L for Pi2, and from 7.89 g/L to 13.23 g/L for Pi4 (Figure 1). The biomass yield of *M. circinelloides* was, in several cases, higher than in the reference medium with Pi1 concentration (Figure 1, Appendix A).

Metal ions affected the growth of *M. circinelloides* differently and the strongest effect was observed at moderate and high levels of phosphates (Pi1, Pi2, and Pi4). Metal ions’ starvation for most of the tested metal ions led to a reduced fungal growth. Thus, Zn and Mg starvation resulted in a low or no growth of *M. ciricinelloides* in all tested Pi conditions (Figure 1, Appendix A). Further, removal of Cu ions led to a slight biomass decrease. Removal and low concentrations of Ca ions resulted in a slight biomass decrease at Pi2 and Pi4 and increase at Pi0.25, Pi0.5, and Pi1 levels (Figure 1, Appendix A). Similar increase in the biomass yield was observed for the media without Fe ions at Pi0.25, and it was significantly decreased at higher Pi levels −44% for Pi1, −30% Pi2, and −16% for Pi4 compared to the reference amount of Fe (Figure 1, Appendix A). Deficiency of Co ions did not result in any significant change in biomass formation of *M. circinelloides* in comparison with high concentrations of Co ions (100Co).

Generally, increased availability of metal ions showed either no effect or both growth-stimulating and -inhibiting effects, depending on the metal type and phosphates concentration. Thus, high metal ions concentration in the media with low Pi levels did not lead to any significant changes in the biomass formation of *M. ciricinelloides* (Figure 1), with the exception of Fe and Zn ions. High amount of Fe ions in a combination with the low concentration of phosphates showed a negative effect on the biomass formation, and no or very limited growth was observed for the following Fe conditions: 1000Fe-Pi 0.25, 1000Fe-Pi1, and 100Fe-Pi0.25. Low growth under these conditions can be connected to the acidic pH ranging from 2.79 for Pi4 to 1.62 for Pi0.25 (Appendix A). The opposite effect was observed for the media with elevated concentration of Zn ions, which enhanced growth of *M. circinelloides*, resulting in a higher biomass yield under Pi limitation, with the highest biomass yield of 4.44 g/L observed for 1000Zn condition (1 g/L ZnSO_4_·7H_2_O) (Figure 1, Appendix A).

A considerable effect of the increased metal ions availability was recorded for the media with moderate and high Pi levels (Figure 1). For example, high concentrations of Mn provided higher biomass at Pi4, with the highest yield of 12.93 g/L for 1000Mn condition, while the effect of other Mn conditions on the growth of M. *circinelloides* was generally negligible. Increasing amount of Fe ions up to the 100Fe condition positively affected the biomass production of *M. circinelloides* in the media with Pi4 and Pi2, and the highest yield of 11.84 g/L was observed for the condition 100Fe with Pi4 (Figure 1). However, very high iron concentration (1000Fe) showed an inhibiting effect at Pi1 and significantly decreased biomass at Pi2 and Pi4 (Figure 1). While increased concentration of Zn ions positively affected the growth of *M. circinelloides* in the media with low phosphates concentration (Pi0.5 and Pi0.25), it slightly decreased the growth in media with high phosphates concentration (Pi2 and Pi4). Generally, it can be concluded that increased concentrations of Zn ions (10Zn and 100Zn) in the media have beneficial effects, since under all Pi levels the biomass yield was increased compared to the standard conditions (R) (Figure 1). A similar effect can be seen for Cu and Co ions, where media with 10Cu condition provided the highest biomass yield for moderate and high Pi levels (Figure 1). Moreover, the highest biomass yield of 13.23 g/L of all tested conditions was observed for 10Co (0.001 g/L CoSO_4_·7H_2_O) with Pi4 level of phosphorus substrate (Figure 1, Appendix A).

### 3.2. Effect of Metal Ions on Lipid Accumulation and Fatty Acid Profile of M. Circinelloides TAGs

Lipid content in oleaginous *M. circinelloides* biomass grown in the different media is reported in Table 3. Due to the low growth and not sufficient amount of biomass for lipid extraction, samples Pi0.25 and Pi0.5 for all metal ions conditions, and samples 1000Fe, 0Mg, 0Zn, 10Ca0Mg were excluded from the lipid extraction and further data analysis.

Lipid accumulation in *M. circinelloides* grown under the reference metal ion conditions reached approximately 41% for Pi1 and 33% for Pi2 and Pi4 (Table 3). Lack of several metals resulted in an increase of lipid content for several Pi conditions. For example, removal of Ca, Co, and Fe ions in the media with Pi2 and Pi1 and Cu and Mn ions in the media with Pi1, Pi2, and Pi4 resulted in an increase in lipid accumulation in *M. circinelloides*, compared to the reference conditions. The most significant increase in lipid accumulation was recorded for 0Ca-Pi1 and 0Cu-Pi1 conditions. Interestingly, the removal of some metal ions, such as Ca, Co, and Fe, enhanced lipid accumulation only at moderate phosphate concentrations in the media (Pi1 and Pi2), and decreased lipid accumulation at high phosphate concentrations (Pi4). Removal of Mn, and especially Cu, ions resulted in increased lipid accumulation at moderate and high phosphate concentrations (Pi1, Pi2, and Pi4) (Table 3).

Variation in the availability of metal ions showed diverse and metal-specific effects on the lipid accumulation in *M. circinelloides* (Table 3). Lack of Mn ions has resulted in relatively high lipid accumulation in *M. circinelloides* for all tested Pi levels. The inhibiting effect of higher Mn ion concentrations was more visible in the media with reference amounts of phosphates (Pi1). Similar results were recorded for the media with increased levels of Fe ions, where lipid yield was lower than under the reference Fe condition (R). Two tested concentrations of Mg ions provided low lipid yield in *M. circinelloides* with the lowest values of 11.43% at 0.01Mg-Pi1 condition, while lipid yield of 39.4% was observed in the biomass grown in the 0.1Mg-Pi4 condition. When increasing the amount of Co ions in the media with Pi4 and Pi2, a decrease in lipid accumulation was recorded, while the opposite effect was seen for the medium with Pi4. An increase in the concentration of Zn ions showed a triggering effect on lipid accumulation in *M. circinelloides* grown at different levels of phosphorus substrate, with the highest lipid yield of 49.78% at 10Zn-Pi1 (0.01 g/L ZnSO_4_·7H_2_O), which was 9% higher than for the reference condition. A similar lipogenesis triggering effect was observed for increasing concentration of Cu ions at all tested Pi levels, while the highest lipid yield was recorded when Cu ions were removed. The most diverse effect on lipid accumulation in *M. circinelloides* was observed for different concentrations of Ca ions. Increase in Ca ion availability from 0Ca to 0.01Ca resulted in the decrease of lipid yield for condition 0.01Ca-Pi1, while a further increase in Ca ions to 0.1Ca resulted in the increase of lipid yield for biomass grown in the media with Pi1, Pi2, and Pi4. A high concentration of Ca ions (10Ca) resulted in the decrease in lipid accumulation in the medium with Pi4 and slight increase in the media with Pi2 and Pi1 (Table 3).

The fatty acid profile of *M. circinelloides* grown under the reference condition was dominated by oleic acid (C18:1n9; 38%), followed by palmitic (C16:0; 22%), linoleic (C18:2n6; 14%), and γ-linolenic (C18:3n6; 12%) acids. Further, stearic (C18:0; 5%), palmitoleic (C16:1; 1.75%), and myristic (C14:0; 1.5%) acids were recorded in smaller amounts (Figure 2, Appendix A). An example chromatogram can be found in the Appendix A. The fatty acid profile of *M. circinelloides,* grown under reference metal ion conditions, slightly changed depending on the phosphorus availability in the media. Thus, we observed an increase in the unsaturation and amount of palmitoleic acid with the increasing amount of phosphorus (Figure 2, Appendix A). An opposite effect of phosphorus availability (and the associated changes in pH of media) was recorded for the unsaturation of stearic acid into oleic and γ-linolenic acid, where decreasing unsaturation was evident with increasing Pi concentrations and higher pH. This pattern can be visible through all the samples, with some exceptions for 10Fe-Pi1/Pi2/Pi4, 100Fe-Pi1/Pi2, 1000Zn, 1000Co, 10Cu-Pi2, and 0.01Mg-Pi1conditions (Figure 2, Appendix A). Minimal content of myristic acid (C14:0) was observed in 10Fe and 100Fe conditions, except for the 100Fe-Pi4 sample (Figure 2). Further, media with high amounts of Zn (1000Zn) and Co (1000Co) ions led to the synthesis of TAGs with the increased relative amount of stearic acid (Figure 2, Appendix A).

To reveal underlying correlations among certain fatty acids, design variables, and reference variables, as well as sum of saturated (SAT), monounsaturated (MUFA), and polyunsaturated (PUFA) fatty acids, PCA analysis of fatty acid (FA) profiles was done for each Pi substrate level separately. The separation of data into different Pi concentrations was done in order to focus on the effect of metal ions only on FA profile, excluding the effect of phosphorus substrate availability. The results, in the form of correlation loading plots, are presented in Figure 3. Generally, the fatty acid profile of *M. circinelloides* was quite consistent, irrespective of media composition.

In the biomass obtained from the media with Pi1 and Pi2 amounts of phosphorus, high concentrations of Co (1000 Co) and Zn (1000 Zn) ions were positively correlated with the saturated fatty acids (SAT) (Figure 3A). This was also evident from the detailed FA profiles, where the relative amount of palmitic and stearic acid was increased under these conditions (Figure 2). Further, some tendency of positive correlation between increasing concentration of Fe ions and content of polyunsaturated fatty acids (PUFA) was observed in the media with Pi1 level (Figure 3A). In the media with high amounts of phosphorus substrate (Pi4 and Pi2), 1000 Mn and 0.01 Mg conditions were positively correlated with the polyunsaturated fatty acids (PUFAs) (Figure 3B,C).

### 3.3. Chemical Composition of M. Circinelloides Biomass

In order to study differences in the compositional profile of the *M. circinelloides* biomass, high-throughput Fourier transform infrared (FTIR-HTS) spectroscopy was used. Spectral regions and peaks related to three types of metabolites—lipids, chitin/chitosan, and polyphosphates were used in the analysis. The FTIR-HTS spectra (Appendix A) showed that fungal biomass was dominated by signals of these intracellular metabolites. The spectra of reference materials can be found in the Appendix A. The maxima of the peaks selected for the correlation loading plots are listed in Table 2. Due to the insufficient growth (Figure 1), the following samples have been disregarded from the FTIR-HTS spectral data analysis: (i) all samples grown under Pi0.25 and Pi0.5 levels; (ii) samples grown in the media with 1000Fe, 0Mg, 0.01Mg, 0Zn, and 10Ca0Mg.

First, we examined the lipid region of FTIR-HTS spectra (3050–2800 and 1800–1700 cm^−1^) and analyzed the correspondence of it with GC data by PCA analysis (Figure 4). On the PCA score plots we can observe similar pattern for FTIR and GC data, indicating similarity in the obtained information about lipid yield and profile from the data of these analytical techniques.

The correlation loading plots from the PCA analysis visualize the relation between the presence of lipids, chitin/chitosan, and polyphosphate and different media (Figure 5). PCA analysis of EMSC preprocessed spectra was performed separately for different Pi concentrations in order to emphasize the effect of different metal ions and disconnect it from the effect of inorganic phosphorus substrate. The loading vectors and FTIR spectral scores are displayed in Appendix A. The first principle component (PC1), which explained the highest variance in the FTIR data, was represented by lipids to proteins and to chitin and chitosan ratio. The second principle component (PC2) was represented by the polyphosphate peaks, which were strongly visible in the cases of polyphosphate accumulation triggering conditions when Pi4 and Pi2 phosphate concentrations were used in the media (Appendix A), while no strong characteristic signals representing any of the studied metabolite were visible in the PC2 for Pi1 condition (Appendix A).

Lipids and chitin/chitosan are both carbon-rich metabolites, therefore their synthesis processes are competing for the C source. In all correlation loading plots, we can observe that lipids and chitin/chitosan were anticorrelated, indicating that these metabolites cannot be produced simultaneously at high yields, while they still can be coproduced with one of them dominating (Figure 5). Further, we see that peaks 2879 cm^−1^ (-C-H stretching) and 950 cm^−1^ (C-O str, C-C str., C-O-H def. C-O-C def), responsible for chitin/chitosan, have been shown to be correlated with the lipid-related peaks (Figure 4). The reason is that the chemical bonds, represented by these peaks, are also present in lipids and the contribution of the lipid associated peaks was stronger than the contribution of chitin/chitosan peaks. The chitin/chitosan formation could also be negatively affected by the N-limitation. The biomass concentration was correlated with the lipid peaks, revealing that a good lipid accumulation can be achieved only with the optimal growth conditions providing good growth and biomass formation and that high biomass concentration was the result of the increased lipid accumulation (Figure 5).

The effect of metal ions on the *M. circinelloides* biomass composition in the media with reference level of phosphorus (Pi1) is displayed in a PCA score plot with loading vectors (Appendix A). Due to the fact that PC2 was not representing any clear relation between FTIR-HTS peaks and studied metabolites, we analyzed only PC1 representing the ratio of lipids to protein and to chitin and chitosan peaks. PCA score plots show that lipid content correlated with the absence of Ca ions (0Ca) (Appendix A), which is in agreement with the lipid yield data presented in Table 3. Further, the PCA score plot shows that increasing Co amount displayed an inhibiting effect on the lipid accumulation (Appendix A). The same results were observed for the lipid yield, where the highest lipid production was recorded for 0Co-Pi1 condition (Table 3). FTIR-HTS spectra of biomass grown in the media with reference level of inorganic phosphorus substrate (Pi1) did not exhibit significant absorbance for polyphosphate peaks. The correlation loading plot (Figure 5A) shows some effect of two metal ions—Fe and Co. Namely, a decrease in the concentration of Fe ions was correlated with the chitin/chitosan peaks, indicating that higher amounts of this metabolites is expected in the fungal biomass when Fe ions are in a low availability, while the absence of Fe ions (0Fe) was anticorrelated with the lipid peaks and biomass concentration (Figure 5A). This corresponds well to the biomass production results, where it is obvious that lack of Fe ions caused a significant decrease of biomass formation (Figure 1). A high concentration of Co ions (1000 Co) showed correlation with chitin/chitosan peaks of FTIR-HTS spectra, indicating that the relative content of this metabolite in the fungal biomass increased with the increased concentration of Co ions. Absence of Ca ions (0Ca) was correlated with the high lipid and biomass concentration. This is in agreement with the reference biomass and lipid concentration results, where 0Ca-Pi1condition provided the highest biomass and lipid production from all the tested Ca ions conditions (Figure 1 and Figure 2).

The effect of metal ions on the synthesis of studied *M. circinelloides* metabolites in the media with Pi2 level of phosphorus is displayed on the Figure 5B, where we can see that increased amounts of Zn and Cu ions were corelated with the lipid yield. Similar correlation results were observed for Co ions, except for the condition of 1000Co-Pi2, which was slightly anticorrelated with lipid peaks. Correlation between polyphosphate spectral peaks and the highest tested Ca ions amount (10Ca) was observed (Figure 4B). Further, we can see correlation between amount of Zn ions and polyphosphate peaks. Finally, metal ions conditions 0.1Ca, 1000Zn, 100Zn, 10Zn, 10Co, 100Co, and 10Cu correlated with the lipid yield.

When examining the effect of metal ions under the high amounts of inorganic phosphorus substrate level (Pi4), on the PCA score plot it is seen that increasing amount of Ca ions corelated with the decrease of the relative lipid content and increase of the polyphosphate content (Appendix A). Further, in Figure 5C we observed that: (i) decreasing Cu ion availability and high concentration of Zn ions correlated with the increase of relative lipid content; (ii) low amount of Ca ions (0,1Ca) correlated with the lipid peaks and lipid yield and anticorrelated with polyphosphates peaks, while high amount of Ca ions (10Ca- 1 g/L CaCl_2._2H_2_O) correlated with polyphosphate and chitin/chitosan peaks; (iii) low concentration of Mg ions (0,1Mg) correlated with lipid peaks and anticorrelates with polyphosphate peaks; (iv) there was no correlation observed for Co and Mn ions (Figure 5C).

## 4. Discussion

For half of a century, *M. circinelloides* has been studied as a microbial cell factory for production of a series of metabolites and valorization of different substrates. Today, this dimorphic oleaginous fungus is positioned as one of the most robust fungal cell factories for the biotech, biorefinery, and bioremediation industries [2].

Despite the deep understanding of *M. circinelloides* physiology and metabolic processes, the role and the effect of metal ions on the lipid accumulation and the cellular composition of this fungus have not been systematically investigated. The effect of metal ions on the growth and metabolic activity of *M. circinelloides* has, to the authors knowledge, only been assessed in the connection to bioremediation abilities of this fungus [11,12,57]. Therefore, in this study we performed an extensive screening of the growth, lipid accumulation, and compositional profile of *M. circinelloides* on 140 different media with variations in the concentrations of metals ions and phosphorus. Lipid accumulation and fatty acid profiles were determined by the GC-FID. The composition of the fungal biomass was investigated by the quantification of lipids, polyphosphates, and chitin/chitosan, as these components previously have been suggested for a coproduction concept involving *M. circinelloides* [43]. For the evaluation of biomass composition, the modern high-throughput analytical technique FTIR spectroscopy was applied. The main advantage of FTIR spectroscopy is that all biochemical components of the sample can be profiled in a single measurement run, without tedious extraction procedures [58,59,60,61,62]. FTIR spectroscopy provides detailed relative quantitative information about different chemical components of the samples and it has been previously utilized for the characterization of lipids [50,51,63], polyphosphates [64], and chitin/chitosan [65,66]. In this study, we have demonstrated that the FTIR analysis as a sole method coupled to multivariate data analysis can be applied for a fast and simple analysis of microbial biomass.

Efficiency of microbial biomass production, the yield of the targeted metabolite(s), and a coproduction potential are important assessment parameters in bioprocess development [19,20]. Biomass production is affected by factors such as pH, temperature, aeration, media composition, and cultivation mode [2]. For example, culture volume and mode of cultivation were reported by Carvalho et al. as factors strongly affecting the final biomass yield of *M. circinelloides* [67]. The reported biomass production of *M. circinelloides,* depending on the culture conditions, varied greatly, from 5 g/L to 20 g/L [10,17,68]. In our previous studies, the biomass concentration for different *M. circinelloides* strains was between 10 g/L and 15 g/L for the cultivations performed in microtiter plates, and 15.8 g/L in bioreactors [54]. The biomass production of the *M. circinelloides* VI04473 strain in this study varied from 0.5 to 13.2 g/L (Figure 1). The standard growth medium containing reference amount of inorganic phosphorus substrate (Pi1) and the reference amounts of metal ions resulted in 9.8 g/L of biomass, significantly lower than in our previous screening study [50]. The reason for the lower biomass production in the reference medium in the present study was probably due to utilization of ammonium sulphate as a nitrogen source, instead of yeast extract as in the previous study [50]. Ammonium sulphate is a pure inorganic source of nitrogen, lacking any additional macro- and micronutrients, vitamins, and growth factors that are present in yeast extract. Buffering capacity of ammonium sulfate is lower than for yeast extract and, as it has been previously reported, the uptake of ammonium ions causes the release of H^+^ by the fungal cells into the media, which accelerates pH lowering [69,70]. Further, possible formation of sulfuric acid during the uptake of ammonium could occur [71]. In addition, formation of organic acids by fungal cells either during exponential or during the stationary growth phase [72] significantly contributes to the acidity of the growth media. Thus, in the media with the low Pi levels we detected considerably low pH and suppressed growth and lipid accumulation in *M. circinelloides* (Figure 1). Acidic pH is a stress factor for many cellular organelles, especially for endoplasmic reticulum (ER), which is connected to protein folding and lipogenesis in fungal cells. It has been previously reported that acidic pH causes ER stress and induces unfolded protein response (UPR). This results in the accumulation of misfolded proteins in the ER and activation of the ER-stress sensor (Ire1p) and ER stress-responsive transcription factor (Hac1p), leading to the inhibition of growth and metabolic activity [73]. In our previous studies, we have showed that acidic pH affects cell wall and increasing chitin/chitosan production in *M. circinelloides* [43]. Cultivation in Duetz-MTPS does not allow continuous adjustment of pH and only start- and end-point measurements are possible, therefore the effect of low phosphorus concentrations was directly linked to drop in pH. Due to the fact, that acidic pH is quite an aggressive stress factor inhibiting fungal growth, the effect of metal ions on the growth and lipid accumulation under low phosphorus substrate availability was difficult to assess. Only two observations could be considered as significant—increase of biomass under higher Zn ion availability and Ca deficiency. Moreover, under Pi conditions lower than the reference (Pi0.5 and Pi0.25), K and Na ions were compensated with KCl and NaCl salts in order to provide the same Na and K amounts as in the reference Pi1 condition [43,44]. It has been reported that chlorides could have negative effect on the mycelium formation of some fungi [74,75]. Moreover, much higher concentrations of Cl (10–15% NaCl) than used in our study (KCl and NaCl in total below 5%) have shown some negative impact on fungal growth [76]. No negative impact of increased Cl^−^ on the biomass and lipid production has been observed when yeast extract was used as N-source [43,44]. Thus, we can hypothesize that in addition to pH-stress, increased Cl^-^ ions possibly negatively impacted the growth under low pH conditions. Therefore, these samples were excluded from further data analysis.

Variation in metal ion availability showed diverse and often metal- and pH-specific effects on biomass production and biomass composition of *M. circinelloides*. Growth of *M. circinelloides* was severely inhibited in media lacking Zn and Mg, indicating that these metal ions are essential for the growth and metabolic activity of the fungus. Inhibition of fungal growth in the media lacking Zn ions can be related to the fact that Zn plays an important role in the regulation of all genes in the eukaryotic cells [31]. Deficiency of Zn is detrimental for the fungal spore germination and further cell proliferation. Our study shows that elevated concentration of Zn ions has a beneficial effect on the biomass formation under phosphorus limitation. Low concentrations of Mg (0.01Mg condition) led to a decrease in biomass production and lipid yield, especially for Pi1 condition, where a lipid content of only 11.43% was reached (Figure 1, Table 3, Appendix A). This can be explained by the fact that magnesium deficiency in eukaryotic cells can result in the decrease of glucose-6-phosphate, total content of phospholipids, and a remarkable decrease in oxygen and substrate delivery to the cells with further concomitant changes in membrane phospholipids, leading to the reduced cell growth, delay in the cell cycle, and metabolic activity [77]. It has been shown that long-term Mg deficiency for yeast may result in distortion of cell division, production of aberrant cell forms, and a decrease in viability that can lead to a delay or change of cell cycle [78]. Therefore, the difference in the FA profile of *M. circinielloides* grown under the Mg deficiency (0.01Mg condition) could be explained by disruption of the cell cycle [79].

In addition to Zn and Mg, Ca and Fe are known to be essential for fungal growth [31,80]. In our study, an absence of Fe ions in the medium suppressed the growth of *M. circinelloides* under conditions of moderate and high phosphorus concentrations. While these metals did not affect lipid accumulation. This is an interesting observation, due to the fact that Fe is an important cofactor of many enzymes, it is essential during DNA synthesis and cleavage, and, thus, Fe deficiency should strongly affect growth and metabolic activity of fungal cells.

An absence of Ca ions affected growth of *M. circinelloides* depending on the phosphorus concentration and associated pH of the growth media. A considerable increase in the biomass production of *M. circinelloides* was observed in the media lacking Ca ions and containing moderate (Pi1) and low concentrations of phosphates (Pi0.5 and Pi0.25). Elevated biomass production under the condition Ca0-Pi1 could be partially explained by the fact that the absence of Ca ions in the medium enhanced lipid yield up to 61% (*w*/*w*). Increase in lipid accumulation with the decrease of concentration of Ca ions was observed also for media containing Pi2 and Pi4 levels of phosphates. Calcium starvation enhancing lipid accumulation in oleaginous microorganisms has been reported for algae [81], where the lipid production was increased by 30% in Ca deprived media. To the authors knowledge, a similar effect of Ca ions deficiency on lipid accumulation has never been reported for oleaginous fungi. Currently, there is no clear understanding of the mechanisms behind Ca deficiency-induced lipid accumulation in oleaginous microorganisms, and the direct link between calcium and lipid accumulation and TAGs synthesis has not been clearly demonstrated yet. Similar observations have been reported for adipocyte cells, where low cellular availability of Ca ions mediated antilipolytic pathways through a calcium-sensing receptor (CaSR), resulting in enhancing of lipid content in adipose tissue [82]. Due to the fact that lipolytic pathways are functionally conserved from mammalian cells to fungi [83], we suggest that Ca deficiency is mediating similar antilipolytic pathways in oleaginous microorganisms. Further, Wang. W.A. et al. [83] showed that Ca ions are important for the basal sensitivity of the sterol sensing mechanism of the sterol response element binding proteins (SREBPs) pathway. Wang W.A. et al. discovered that reduction of Ca concentration in endoplasmic reticulum changes the distribution of intracellular sterol/cholesterol, resulting in the enhancement of SREBPs activation and triggering synthesis of neutral lipids. Sterol response element binding proteins (SREBP) are transcription factors that are synthesized on endoplasmic reticulum (ER) and considered as ER-associated integral membrane proteins [83]. SREBP were reported for eukaryotic cells, including mammalian and fungal cells [84]. The studies show that SREBP are involved in lipid homeostasis, while SREBP isoforms control the expression of genes responsible for the biosynthesis of sterol/cholesterol, fatty acids, triacylglycerols, and phospholipid in the cell [85]. Further, detailed studies would be needed to confirm if these two events are valid also for oleaginous fungi grown under calcium deficiency.

Increase in *M. circinelloides* biomass yield was observed also at high concentrations of Ca ions in the media with high phosphate concentrations (Pi2). Infrared spectra of *M. circinelloides* biomass grown in this medium showed strong absorbance values for polyphosphate peaks (Appendix A). Thus, we can assume that increase in biomass production is associated with the intracellular accumulation of available inorganic phosphorus substrate in the form of polyphosphate. It has been previously reported that, in media with excess phosphorus source, *M. circinelloides* is able to perform so called luxury uptake of phosphorus and accumulate it in the form of polyphosphates either in the cell wall or in the form of intracellular polyphosphate granules [6]. Polyphosphate (polyP) is a polyanionic compound, and it has been reported by Kikuchi Y. et al. that in the fully dissociated form, polyP has one negative charge per Pi residue and two extra charges of terminal residues [85]. Therefore, accumulation of polyP in the cell results in the accumulation of a large amount of negative charge, which is probably compensated by an existence of a regulatory mechanism for maintaining charge neutrality of the cell. The studies involving temporal and quantitative analyses of cationic components of the fungal cells revealed that Na, K, Ca, and Mg ions were taken up by polyP, providing strong evidence that these ions play a major role in the neutralization of the negative charge of polyP in the fungal cell [85,86]. Thus, it is likely that with the higher availability of calcium ions in the medium, the neutralization of the polyP negative charge is more efficient and a higher amount of phosphorus can be stored intracellularly in the form of polyP. Due to fact that polyphosphate accumulation takes place in the exponential growth phase, while lipid accumulation in the stationary growth phase [87,88], it could be possible to perform a coproduction of these two components by manipulation of the availability of calcium and phosphorus substrate in the medium. Therefore, *M. circinelloides* can be utilized in the phosphorus recycling processes.

In addition to Ca-deprived media, lack of Cu and higher amounts of Zn and Mn considerably enhanced lipid accumulation in *M. circinelloides*. While elevated lipid production observed due to Ca deficiency could be explained by the above-mentioned hypothesis, there is no clear explanation of the high lipid accumulation under the copper deficiency condition that was significantly higher at all Pi levels. It has to be noted that the highest lipid yield was obtained under deficiency of Ca and Cu ions. In the literature, there has only been only one study, conducted on the liver cells, reporting Cu deficiency enhancing lipid storage [89], while metabolic pathways linking copper to lipid homeostasis have not been reported for fungal and any other microbial cells.

The FA profile of the accumulated in *M. circinelloides* TAGs was not significantly affected by the availability of metal ions and phosphorus. Only some tendency in increase of saturation with high Co and Zn amount was observed, but further enzymatic study would be needed to assess the activity of desaturases at these conditions.

By applying FTIR spectroscopy, we revealed that Ca, Co, and Zn ions at different concentrations correlated with lipid peaks; Ca and Zn correlated with polyphosphate, while Fe and Co with chitin/chitosan peaks of *M. circinelloides* biomass spectra. Thus, these ions could be considered as important components in optimizing and developing coproduction of lipids, polyphosphate, and chitin/chitosan by *M. circineloides.* However, further studies are needed to fully understand the role of these metal ions in the metabolic pathways of *M. circineloides* metabolites.

## 5. Conclusions

The aim of the study was to evaluate the effect of different metal ions and their concentration on biomass production, composition, and the lipid production in the oleaginous fungus *M. circinelloides*. Moreover, the growth experiments were conducted at different concentrations of phosphates. It can be concluded that, among tested metals, Mg and Zn are essential metals required for the optimal growth of *M. circinelloides*. Calcium availability is important for optimizing polyphosphate accumulation, while calcium and copper deficiency is important for lipid accumulation in *M. circinelloides*. Tested metal ions did not affect fatty acid profile of the accumulated TAGs. However, Ca, Co, Mg, and Zn ions have affected the cellular biochemical profile of *M. circinelloides*. Thus, metal ions are an important tool for optimizing lipid accumulation and coproduction of lipids, polyphosphate, and chitin/chitosan in *M. circinelloides*.

## Figures and Tables

**Figure 1 jof-06-00260-f001:**
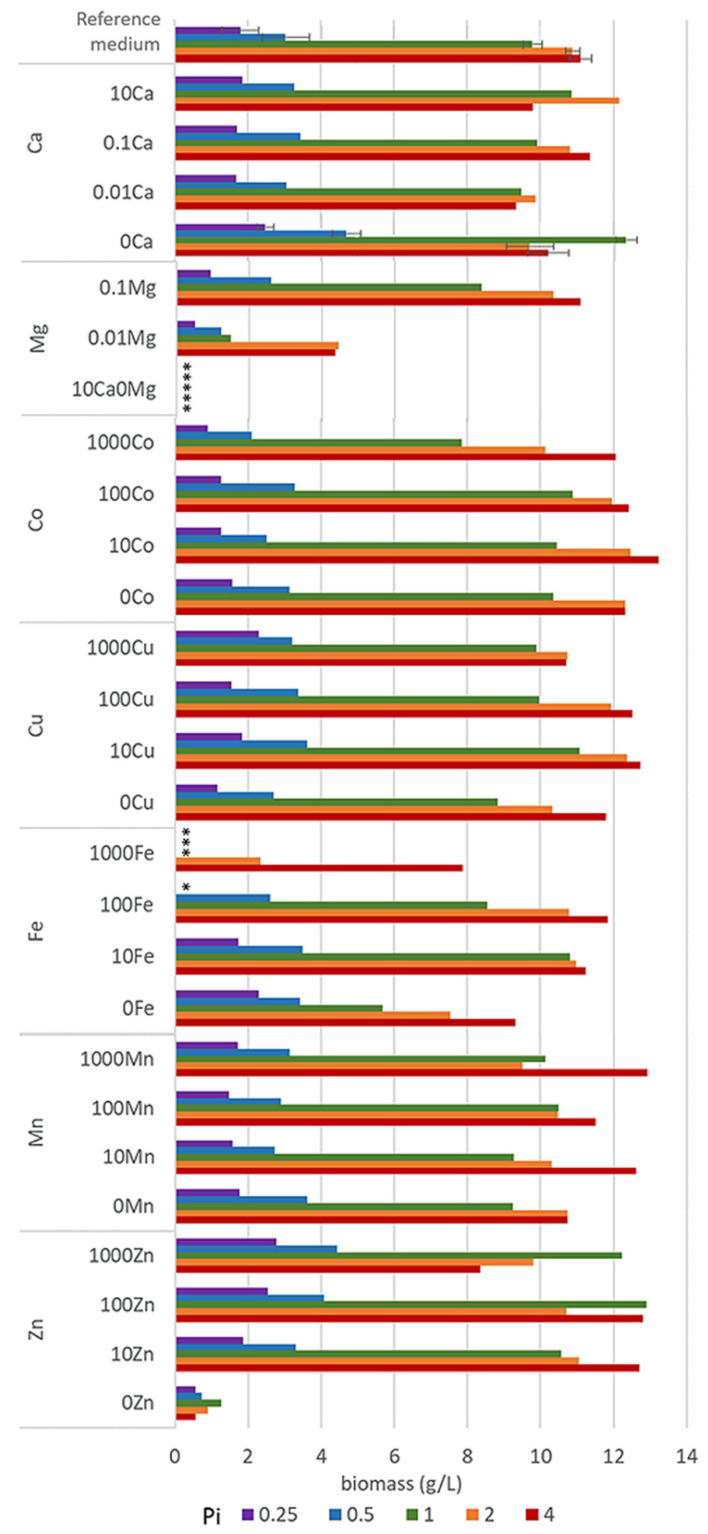
Final biomass concentrations after 7 days of incubation of *M. circinelloides* in the media with different concentrations of metal and phosphorus ions. * Empty slots indicate no growth.

**Figure 2 jof-06-00260-f002:**
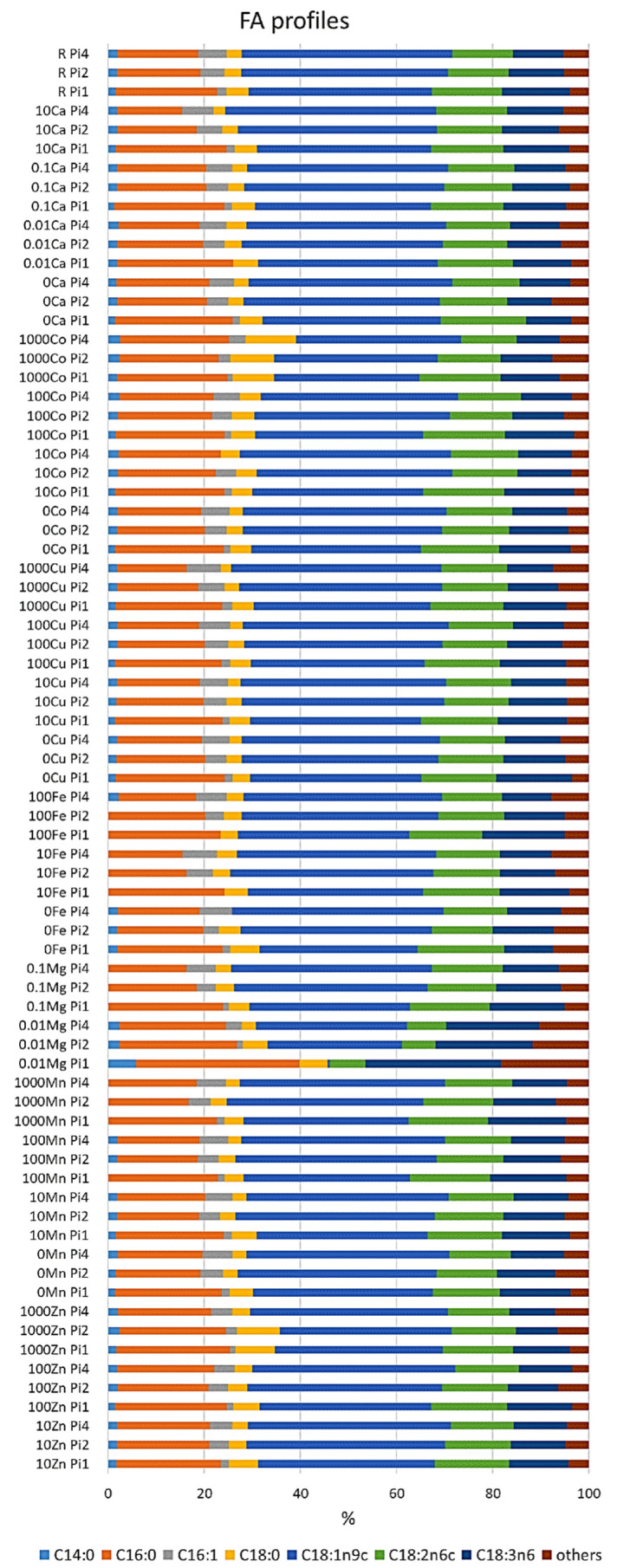
Fatty acid profile of lipids accumulated in *M. circinelloides* grown in media with Pi1, Pi2, and Pi4 levels of phosphorus. Only fatty acids present in amounts of more than 1% are displayed. The rest is summed up as ‘others’. An example chromatogram can be found in the Appendix A.

**Figure 3 jof-06-00260-f003:**
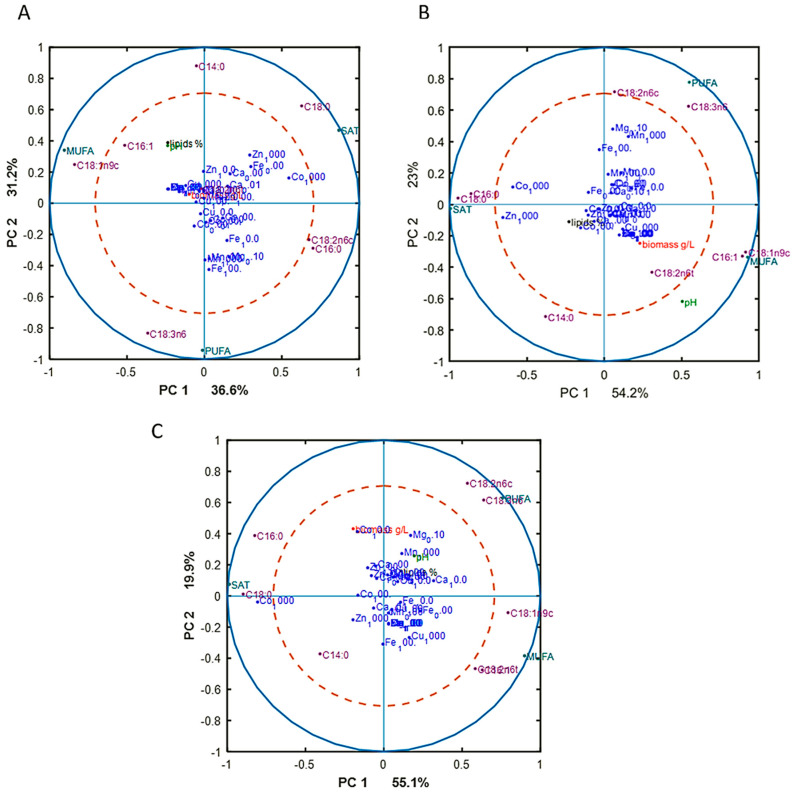
Correlation loading plots based on the PCA analysis of fatty acid (FA) profiles of lipids accumulated in *M. circinelloides* grown in metal ion-regulated media under Pi1 (**A**), Pi2 (**B**), and Pi4 (**C**) levels of phosphorus substrate.

**Figure 4 jof-06-00260-f004:**
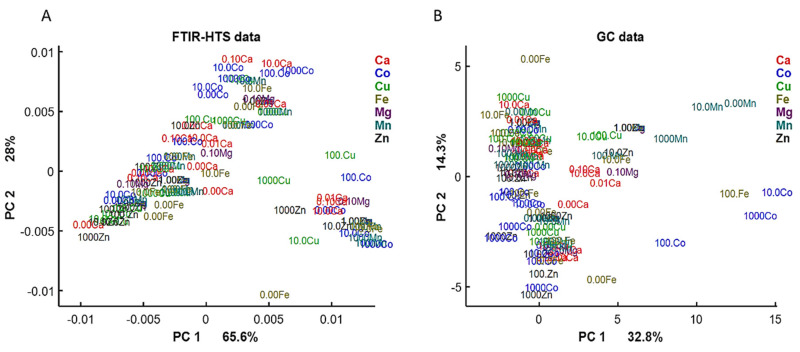
PCA score plots of FTIR-HTS (**A**) and GC (**B**) data. PCA analysis was performed on the preprocessed FTIR-HTS data (2nd derivative, polynomial order 2, window size 13; SROI: 3050–2800 and 1800–1700 cm^−1^, EMSC) and normalized GC data.

**Figure 5 jof-06-00260-f005:**
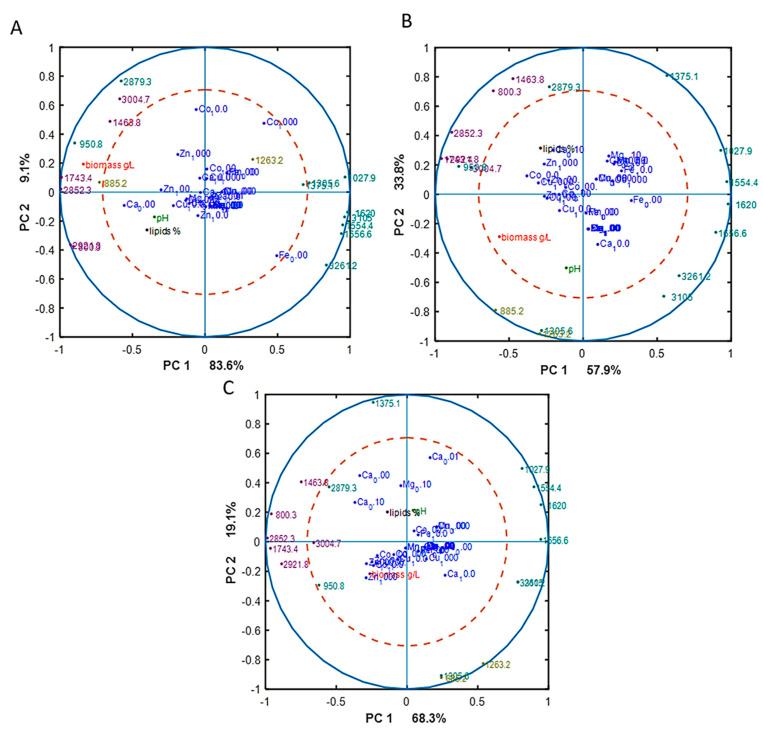
Correlation loading plots based on the PCA analysis of FTIR-HTS spectra of *M. circinelloides* biomass grown in metal ion-regulated media with Pi1 (**A**), Pi2 (**B**), and Pi4 (**C**) levels. Main peaks corresponding to the lipids (purple), chitin/chitosan (green), and polyphosphates (yellow) are presented.

**Table 1 jof-06-00260-t001:** Overview over the relative levels of concentration of metal ions and inorganic phosphate in the media. The exact concentrations can be found in the Appendix A (Appendix A).

Ca	Mg	Cu	Co	Fe	Mn	Zn	Pi
0 0.01 0.1 R 10	0Mg 10Ca 0.01 0.1 R	0 R 10 100 1000	0 R 10 100 1000	0 R 10 100 1000	0 R 10 100 1000	0 R 10 100 1000	0.25 0.5 Pi1 2 4

**Table 2 jof-06-00260-t002:** The maxima of the corresponding chemical bonds selected for correlation loading plots based on FTIR data.

Cell Component	Peak Maxima	Molecular Vibration
Chitin/chitosan	3261	N-H stretching
	3105	N-H stretching
	2879	-C-H stretching
	1656	-C=O stretching (Amide I)
	1620	-C=O stretching (Amide I)
	1554	C-N-H deformation (Amide II)
	1375	-CH_3_ deformation
	1305	C-N-H deformation (Amide III)
	1027	C-O-C str., C-O-H def. C-O-C def.
	950	-CH_3_ def.
Lipids	3004	=C-H stretching
	2921	-C-H stretching
	2852	-C-H stretching
	1743	-C=O stretching
	1463	-CH_2_ bending
	723	>CH_2_ rocking
Polyphosphates	1263	P=O stretching
	885	P-O-P stretching

**Table 3 jof-06-00260-t003:** Lipid accumulation (% of lipids per dry cell weight) for *M. circinelloides* grown in nitrogen-limited metal ion-regulated media with different amounts of inorganic phosphorus substrates (Pi1, Pi2, and Pi4).

Metal Ion Condition	Pi1	Pi2	Pi4
Reference medium	41.13 ± 1.19	33.44 ± 1.28	33.15 ± 0.01
0Ca	61.16 ± 0.16	40.15 ± 2.31	31.51
0.01Ca	34.00	39.61	34.93
0.1Ca	60.55	37.22	43.70
10Ca	44.37	33.50	27.95
0.01Mg	11.43	20.57	22.80
0.1Mg	30.38	32.90	39.40
0Co	38.78	37.24	31.46
10Co	30.25	34.40	29.87
100Co	30.31	38.52	29.60
1000Co	31.49	35.08	29.38
0Cu	61.27	53.80	52.24
10Cu	47.11	37.75	35.63
100Cu	46.70	41.27	38.27
1000Cu	43.11	42.46	38.36
0Fe	37.27	37.00	30.62
10Fe	36.73	34.36	29.19
100Fe	30.77	33.00	27.11
0Mn	46.78	37.94	38.74
10Mn	34.52	31.67	33.21
100Mn	35.16	39.13	34.22
1000Mn	33.23	30.58	33.61
10Zn	49.78	38.31	37.65
100Zn	43.84	41.94	34.72
1000Zn	42.36	41.85	38.04

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
