# Peer review of "Metal and Phosphate Ions Show Remarkable Influence on the Biomass Production and Lipid Accumulation in Oleaginous *Mucor circinelloides"

_jof, 2020, doi:10.3390/jof6040260_

Round 1
Reviewer 1 Report
The manuscript entitled "Influence of metal and phosphate ions on biomass production and lipid accumulation of Mucor circinelloides" provide very complex overview on metabolism of M. circinelloides under various conditions with focus on metal presences.
In my opinion, the manuscript is very well written and I do not have any significant comments or questions. However, there are details that I missed:
1) what was the concentration of spore suspension used for inoculation of media?
2) how do the authors quantified the biomass growth?
Also, the Figures 1 and 2 - especially Fig.2 is for me almost impossible to read. I strongly recommend to display the results as Tables!
Author Response
Reviewer 1
The manuscript entitled "Influence of metal and phosphate ions on biomass production and lipid accumulation of Mucor circinelloides" provide very complex overview on metabolism of M. circinelloides under various conditions with focus on metal presences.
In my opinion, the manuscript is very well written and I do not have any significant comments or questions. However, there are details that I missed:
1) what was the concentration of spore suspension used for inoculation of media?
Authors’ answer: The exact spore concentration in the spore suspension was not estimated. The spore suspensions for all inoculations were prepared in the same way as in the previous studies (Kosa et al. 2017,2018a,2018b). Fresh spore inoculum was prepared for each of 11 biological replicates of reference condition Pi1-R, where the growth showed to be reproducible. Spores were collected from agar plates using 10 ml of saline solution and bacteriological loop, and 7 ml of growth media was inoculated with 50 μl of spore suspension. To provide more details concerning the preparation of spore suspension, we added: ’Spores were collected from agar plates using 10 ml of saline solution and bacteriological loop.’ (see line 103-104).
2) how do the authors quantified the biomass growth?
Authors’ answer: The fungal biomass was separated from the growth media by centrifugation and washed with distilled water. The biomass was freezedried for 24 h and the mass of dry cells measured. See lines 176-177.
Also, the Figures 1 and 2 - especially Fig.2 is for me almost impossible to read. I strongly recommend to display the results as Tables!
Authors’ answer: We agree with the reviewer that a lot of information is contained in Figures 1 and 2. Therefore, according to the Reviewer’s suggestion, tables with biomass concentrations and lipid profiles have been added to the Supplementary Materials (Table S3, Table S4).
Authors would prefer to keep the present graphical representation of the data in the manuscript since the main patterns are still relatively easy to observe in the figures. In case of Figure 2 which shows the fatty acid (FA) profiles, the FA profiles are quite consistent irrespective the substrate composition and changes in FA profiles are easily detectable in the Figure 2.
Kosa, G.; Kohler, A.; Tafintseva, V.; Zimmermann, B.; Forfang, K.; Afseth, N.K.; Tzimorotas, D.; Vuoristo, K.S.; Horn, S.J.; Mounier, J.; Shapaval, V. Microtiter plate cultivation of oleaginous fungi and monitoring of lipogenesis by high-throughput FTIR spectroscopy. Microbial cell factories 2017, 16, 101.
Kosa, G.; Zimmermann, B.; Kohler, A.; Ekeberg, D.; Afseth, N.K.; Mounier, J.; Shapaval, V. High-throughput screening of Mucoromycota fungi for production of low-and high-value lipids. Biotechnology for biofuels 2018, 11, 66.
Kosa, G.; Vuoristo, K.S.; Horn, S.J.; Zimmermann, B.; Afseth, N.K.; Kohler, A.; Shapaval, V. Assessment of the scalability of a microtiter plate system for screening of oleaginous microorganisms. Applied microbiology and biotechnology 2018, 102, 4915-4925.
Reviewer 2 Report
One of the most promising organisms for lipid production is the zygomycete fungus Mucor circinelloides. The authors take up an interesting topic, the influence of metals and phosphate ions on Mucor circinelloides growth with the use Fourier transform infrared (FTIR) spectroscopy. The work is in line with the objectives of the journal. The study though provide interesting novel finding there is some scope for improvement. The following were few suggestion to improve the manuscript.
What was the flow of helium through the column
Could authors add an example of GC-FID chromatogram in supplementary
How the authors measured the total lipids of the fungus for Table 3
According to authors the lipids of M circinelloides are mainly in the form of triacylglycerides, while according to Vicente et al. (2009) polar lipids (phospholipids, sphingolipids and saccharolipids) are large part of the fungal lipids.
Do the authors know the proportions for individual lipid classes for the tested microorganism?
Author Response
Reviewer 2
One of the most promising organisms for lipid production is the zygomycete fungus Mucor circinelloides. The authors take up an interesting topic, the influence of metals and phosphate ions on Mucor circinelloides growth with the use Fourier transform infrared (FTIR) spectroscopy. The work is in line with the objectives of the journal. The study though provide interesting novel finding there is some scope for improvement. The following were few suggestion to improve the manuscript.
1) What was the flow of helium through the column
Authors’ answer: The information about the flow of helium through the column was added in line 163-164:’The flow of helium through the column was 1 mL/min.’
2) Could authors add an example of GC-FID chromatogram in supplementary
Authors’ answer: An example of GC-FID chromatogram has been added in the supplementary materials. See Figure S1.
3) How the authors measured the total lipids of the fungus for Table 3
Authors’ answer: We agree with the reviewer that the total lipid estimation has not been sufficiently described in the materials and methods section of the manuscript. We modified the name of chapter 2.2: ‘Lipid extraction and GC-FID analysis of lipid concentration and fatty acid profile’ by inserting ‘lipid concentration’ to make it clear that GC has been used for the lipid quantification. To provide more detailed information about the lipid quantification, we added following description in lines 168-170: ‘The weight of individual FAs was calculated based on the peak areas, relative response factors (RRF) and C13 internal standard. The total lipids in the fungal biomass are the sum of FA (the weight of C13 IS is subtracted) divided by the weight of dry biomass.’
4) According to authors the lipids of M circinelloides are mainly in the form of triacylglycerides, while according to Vicente et al. (2009) polar lipids (phospholipids, sphingolipids and saccharolipids) are large part of the fungal lipids. Do the authors know the proportions for individual lipid classes for the tested microorganism?
Authors’ answer: Vincente et al. (2009) has been growing M. circinelloides in following way: ‘To produce biodiesel, M. circinelloides biomass was obtained from the prototrophic strain MU241 grown on a solid minimal medium containing glucose as a carbon source (10 g l−1). After three days of growth, a 3.73 ± 0.27 g l−1 of fungal dry mass was obtained. Lipids were extracted from this fungal biomass using three mixtures of solvents: chloroform:methanol (C:M), chloroform:methanol:water (C:M:W) and n-hexane.’
Vincente et al (2009) did not use lipid accumulation triggering medium in their study. Glucose concentration of 10g/L and Yeast Nitrogen Base (YNB) containing approx. 5g/L of ammonium sulphate result in approx. C/N 3.8. Such medium doesn’t provide the N- limiting condition which is necessary for triggering lipid accumulation as it was in our study, where C/N100 was used. Therefore, in the Vincente et al (2009) study, the obtained biomass concentration was 3.73 g/L with 19.9 % (w/w) lipids as a maximum which is quite low compared to our study, where we obtained approx. 40% of lipids (example for ref. condition). In addition, in the Vincente et al (2009) study the cultivation time was shorter and solid media were used. All above mentioned factors considerably contribute to the difference in the lipid profile of M. circinelloides in Vincente et al (2009) and our study.
The lipid profile of M. circinelloides grown under the lipid accumulation triggering condition have been assessed in our previous study by thin layer chromatography (Forfang et. al. 2017) and the following lipid classes were identified: 70 %TAG, 10.7 % mono- di- glycerides, 10.1% free FA, 9.2% phospholipids.
Forfang, K.; Zimmermann, B.; Kosa, G.; Kohler, A.; Shapaval, V. FTIR spectroscopy for evaluation and monitoring of lipid extraction efficiency for oleaginous fungi. PloS one 2017, 12, e0170611.
Vicente, G.; Bautista, L.F.; Rodríguez, R.; Gutiérrez, F.J.; Sádaba, I.; Ruiz-Vázquez, R.M.; Torres-Martínez, S.; Garre, V. Biodiesel production from biomass of an oleaginous fungus. Biochemical Engineering Journal 2009, 48, 22-27.
Reviewer 3 Report
Influence of metal and phosphate ions on biomass 3 production and lipid accumulation of Mucor circinelloides by Simona Dzurendova is a work that presents interesting results
The work must be corrected before its acceptance by the Journal of Fungi (MDPI).
The title should be reformulated to highlight the importance of the work.
The scientific name of the microorganism object of the work "Mucor circinelloides" is often abbreviated and sometimes not, please unify a criterion so that the presentation is correct. Once named the first time, M. Mucor circinelloides can be used. Always in cursive letters. Please review the entire document.
Line 14: Mucor circinelloides
Line 41: M. circinelloides
Line 44: M. circinelloides
The names of the metals used must be indicated the first time magnesium (Mg) is used ...
Check that all abbreviations are correctly defined to facilitate reading
(NH4) 2SO4 1.5, KH2PO4 7, Na2HPO4 2, MgSO4 · 7H2O 1.5. The concentration tested was g / L. It is not clear. In the tables the concentration of the metals should be indicated there is also not clear.
Line 106: FeCl3 · 6H2O 0.008, the concentration 0.008 g / L of FeCl3 · 6H2O should be indicated first ... correct throughout the document because this makes reading difficult
Line 118: 0Mg10Ca which… this format is used throughout the text, please indicate the concentrations throughout the text and check all the tables
Line 147: 90 ° C for 1 hour. Change for 90 ° C for 1 h correct throughout the manuscript
Line 162: 70 ° C for 2 minutes, after 8 minutes change for 70 ° C for 2 min, after 8 min. Correct the entire manuscript
Line 243: and from 0.73 g / L to 4.7 g / L, respectively (Figure 1). Unify criteria in the number of decimal used
Table 3: 41.13 ± 1.19 33.44 ± 1.28 33.15 ± 0.01 please change (,) to (.) Use the international measurement system. Reviewing the entire manuscript is a recurring error in the text
Figure 2 is very complex to analyze, please synthesize its content or present the results in another way
Rodrigues Reis, C.E .; Bento, H.B .; Carvalho, A.K .; Rajendran, A .; Hu, B .; De Castro, H.F. Critical applications of Mucor circinelloides within a biorefinery context. Critical reviews in biotechnology 2019, 39, 555-570. Review the names of the journal Critical Reviews in Biotechnology.
Author Response
Reviewer 3
Influence of metal and phosphate ions on biomass 3 production and lipid accumulation of Mucor circinelloides by Simona Dzurendova is a work that presents interesting results
The work must be corrected before its acceptance by the Journal of Fungi (MDPI).
1) The title should be reformulated to highlight the importance of the work.
Authors’ answer: The title of the manuscript was modified:
Metal and phosphate ions show remarkable influence on the biomass production and lipid accumulation in oleaginous Mucor circinelloides.
2) The scientific name of the microorganism object of the work "Mucor circinelloides" is often abbreviated and sometimes not, please unify a criterion so that the presentation is correct. Once named the first time, M. Mucor circinelloides can be used. Always in cursive letters. Please review the entire document.
Line 14: Mucor circinelloides
Line 41: M. circinelloides
Line 44: M. circinelloides
Authors’ answer: The abbreviation ‘M. circinelloides’ have been used throughout the manuscript instead of ‘Mucor circinelloides’.
3) The names of the metals used must be indicated the first time magnesium (Mg) is used ...
Check that all abbreviations are correctly defined to facilitate reading
Authors’ answer: The abbreviations of the metals were added the first time mentioned in the introduction (see lines 58, 59, 61, 62, 71, 73, 77).
4) (NH4) 2SO4 1.5, KH2PO4 7, Na2HPO4 2, MgSO4 · 7H2O 1.5. The concentration tested was g / L. It is not clear. In the tables the concentration of the metals should be indicated there is also not clear.
Line 106: FeCl3 · 6H2O 0.008, the concentration 0.008 g / L of FeCl3 · 6H2O should be indicated first ... correct throughout the document because this makes reading difficult
Authors’ answer: The text explaining the reference media composition has been modified according to the Reviewer’s suggestion:’… is the following: 80 g/L of glucose, 1.5 g/L of (NH4)2SO4, 7 g/L of KH2PO4, 2 g/L of Na2HPO4, 1.5 g/L of MgSO4·7H2O, 0.1 g/L of CaCl2·2H2O, 0.008 g/L of FeCl3·6H2O, 0.001 g/L of ZnSO4·7H2O, 0.0001 g/L of CoSO4·7H2O, 0.0001 g/L of CuSO4·5H2O, 0.0001 g/L of MnSO4·5H2O,...’ (see lines 106-108). Table 1 contains the overview of samples with the markings explaining the modification of the concentrations throughout the experiments and, thus, providing the overview about the experimental design, while concrete concentrations are listed in the Table S1.
5) Line 118: 0Mg10Ca which… this format is used throughout the text, please indicate the concentrations throughout the text and check all the tables
Authors’ answer: The abbreviations, such as 0Mg10Ca were used in order to make the manuscript easy to read. Also, to make it easy to find the correspondence between the figures and samples mentioned in the text. The concrete concentrations have been provided in the materials and methods and the Table S1 and we refer to it in the text (see line 122). However, we agree with the Reviewer that concrete concentrations are useful information and we included it with the most important results throughout the manuscript text (see lines 270, 287, 320, 451).
6) Line 147: 90 ° C for 1 hour. Change for 90 ° C for 1 h correct throughout the manuscript
Authors’ answer: The word hour has been replaced with the abbreviation ‘h’ (See lines 149, 177, 180).
7) Line 162: 70 ° C for 2 minutes, after 8 minutes change for 70 ° C for 2 min, after 8 min. Correct the entire manuscript
Authors’ answer: The word minutes has been replaced with the abbreviation ‘min’ (See lines 164, 165, 166).
8) Line 243: and from 0.73 g / L to 4.7 g / L, respectively (Figure 1). Unify criteria in the number of decimal used
Authors’ answer: The criteria in number of decimals have been unified (see line 243).
9) Table 3: 41.13 ± 1.19 33.44 ± 1.28 33.15 ± 0.01 please change (,) to (.) Use the international measurement system. Reviewing the entire manuscript is a recurring error in the text
Authors’ answer: Commas have been replaced by dots in Table 3. The entire manuscript has been reviewed.
10) Figure 2 is very complex to analyze, please synthesize its content or present the results in another way
Authors’ answer: We agree with the Reviewer that Figure 2 contains a lot of information. In general, fatty acid profiles showed to be quite consistent and changed very little under the different metal ions and phosphorus concentrations. Thus, we believe that a graphical representation of these results (Figure 2) provides a satisfactory overview over the results and we would like to keep the figure in the manuscript. But in order to simplify the reading of Figure 2, we added a table with the fatty acid profiles to the Supplementary Materials (See Table S4).
11) Rodrigues Reis, C.E .; Bento, H.B .; Carvalho, A.K .; Rajendran, A .; Hu, B .; De Castro, H.F. Critical applications of Mucor circinelloides within a biorefinery context. Critical reviews in biotechnology 2019, 39, 555-570. Review the names of the journal Critical Reviews in Biotechnology.
Authors’ answer: The name of the Journal has been reviewed and capitalized.
Round 2
Reviewer 1 Report
I have no other comments.
Reviewer 2 Report
The authors have addressed all my concerns. I am recommended to accept the manuscript in the current form for publication.
Reviewer 3 Report
Thanks to the authors for accepting suggestions to improve the manuscript. I think that the current form of the article is correct to be accepted by JOF. Best regards.
A small comment: you should disable track changes in the document to make the text easier to read. Thus the presentation of the article is more appropriate.